# InfoCons: Identifying Interpretable Critical Concepts in Point Clouds via Information Theory

**Feifei Li** [1]  **Mi Zhang** [1]  **Zhaoxiang Wang** [1]  **Min Yang** [1]

## Abstract

Interpretability of point cloud (PC) models becomes imperative given their deployment in safety-critical scenarios such as autonomous vehicles. We focus on attributing PC model outputs to interpretable critical concepts, defined as meaningful subsets of the input point cloud. To enable human-understandable diagnostics of model failures, an ideal critical subset should be *faithful* (preserving points that causally influence predictions) and *conceptually coherent* (forming semantically meaningful structures that align with human perception). We propose InfoCons, an explanation framework that applies information-theoretic principles to decompose the point cloud into 3D concepts, enabling the examination of their causal effect on model predictions with learnable priors. We evaluate InfoCons on synthetic datasets for classification, comparing it qualitatively and quantitatively with four baselines. We further demonstrate its scalability and flexibility on two real-world datasets and in two applications that utilize critical scores of PC.

## 1. Introduction

Point clouds are unordered sets of points representing the 3D world. Point cloud (PC) models directly take the point cloud as input and perform various downstream tasks such as classification, segmentation and object detection (Chang et al., 2015; Armeni et al., 2016). The advancements in deep learning-based PC models (Qi et al., 2017a;b; Wang et al., 2019; Guo et al., 2021; Ma et al., 2022; Wu et al., 2024) have significantly improved their performance on 3D understanding tasks, paving the way for applications in

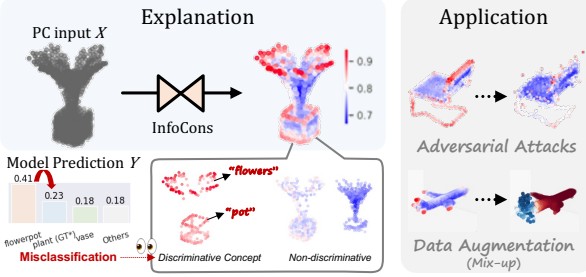

*Figure 1.* Attributing PC model outputs to a group of interpretable critical concepts using InfoCons. The derived concepts, which conform to a specific semantically meaningful structure (i.e., conceptual cohesion), can reflect their influence on the model outputs faithfully. The critical score map can also be integrated into multiple applications. Better viewed in color.

safety-critical scenarios like autonomous vehicles (Geiger et al., 2012). Interpretability has become a crucial issue in assessing the reliability of model decisions, and understanding complex, high-dimensional deep neural networks has been a long-standing challenge. However, despite substantial recent progress in the domains of 2D images and language (Deng et al., 2024; Rai et al., 2024), research in the field of 3D data remains lacking. Challenges in developing explanation methods for point clouds arise from the significant domain gap between point clouds and other modalities, making direct application of 2D methods less effective (Wu et al., 2015; Tan, 2023). Furthermore, the absence of well-established architectures and diverse modules complicates the development of a faithful and broadly effective explanation method for various point cloud models.

Previous study of point cloud explanation methods aim to extract a point cloud subset critical for model decisions, collectively referred to as *Critical Subset Theory* (Qi et al., 2017a; Zheng et al., 2019; Kim et al., 2021; Levi & Gilboa, 2023). These methods differ in how they define the importance of points. For instance, the maxpool-based *Critical Points (CP)* (Qi et al., 2017a) identify points *indexed* through a maxpooling layer applied to features as critical, suggesting that *stronger responses* indicate greater importance. Meanwhile, the gradient-based *point cloud saliency map (PCSAM)* (Zheng et al., 2019) utilizes the gradients of the CE loss *w.r.t.* spherical coordinates to rank salient

[1]School of Computer Science, Fudan University, China. Correspondence to: Mi Zhang <mi_zhang@fudan.edu.cn>, Min Yang <m_yang@fudan.edu.cn>.

points, indicating that *sensitivity* to a target label reflects importance. Critical subsets can naturally explain model decisions in an interpretable way, mitigating the issue of information redundancy, which arises from the *large number of points* representing the same amount of information.

However, existing methods have limitations in providing interpretable subsets that satisfy both faithfulness and conceptual cohesion. As theoretically illustrated in Fig. 2-(a), faithfulness means that the subset has a direct causal effect on *model predictions*, thereby providing sufficient information about the model's behavior in cases of misclassification (e.g., why the model confuses 'plant' with 'flower_pot' as shown in Fig. 1). Meanwhile, conceptual cohesion encourages the subset to conform to a semantically meaningful structure that aligns with human perceptual priors (e.g., object parts like *flower* and *pot/vase* that can be identified as contributing to the misclassification). From this perspective, existing methods fail to satisfy both criteria. For example, the maxpool-based CP, which relies on the output of the PC encoder, neglects the influence of the subsequent classifier, resulting in a lack of faithfulness to the entire PC model. Moreover, PCSAM approximates points removal by slightly perturbing points towards the PC centroid, introducing a biased shape prior of *centroid singularity*[1]. This prior leads to biased attribution, producing critical subsets that are spatially concentrated in corners.

In our work, we propose a novel framework, InfoCons, that applies information-theoretic principles to decompose a point cloud into 3D concepts with different levels of influence on model predictions, where the *critical subset* is given by the most discriminative concept. To ensure faithfulness, we leverage *mutual information (MI)* between the critical subset $C$ and the model decision $Y$, deriving a critical score map by maximizing $I(C, Y)$. Meanwhile, we introduce a learnable unbiased prior to minimize the MI between the critical subset and input point cloud $X$, encouraging a meaningful conceptual structure. Further motivated by variational *Information Bottleneck (IB) for Attribution* frameworks, which aim to select informative features as model interpretation via a learned selective function (Chen et al., 2018; Schulz et al., 2020; Bang et al., 2021), we formalize the learnable prior as a selective function for critical subsets, optimized using an IB-based objective. The proposed framework for explaining PC models specifically addresses the problem of selecting the most influential subsets from an unordered set of points. This contrasts with existing methods for CNNs on 2D images, which focus on selecting relevant pixels of the labeled objects from *irrelevant* background pixels.

---

[1]Due to the inductive bias of PC models, a PC of singularity ($N = 1024$ points at the same position) is also shape-informative: an 'origin PC' can be predicted as '*guitar*' by PointNet (Qi et al., 2017a) with 99.2% certainty.

We conduct both qualitative and quantitative evaluations of InfoCons using the synthesis dataset ModelNet40 (Chang et al., 2015), explaining eight models covering three types of structures: non-hierarchical MLP-based, hierarchical MLP-based, and self-attention-based. We apply InfoCons to the saliency-guided mixup method SageMix (Lee et al., 2022) and the sensitivity-guided attack SIAdv (Huang et al., 2022) to further enhance their performance. We also demonstrate the scalability of InfoCons on more challenging benchmarks, including the real-world object dataset ScanObjectNN (Uy et al., 2019) and outdoor scene dataset KITTI (Geiger et al., 2012).

In summary, our main contributions are as follows:

- We formalize the problem of extracting interpretable critical subsets for deep PC models and provide an in-depth analysis on the limitations of existing methods in terms of *faithfulness* and *conceptual cohesion*.

- We propose a novel framework, InfoCons, which applies information-theoretic principles to decompose the point cloud into 3D concepts by learning a selective function with unbiased prior.

- We conduct comprehensive experiments across three datasets, evaluating InfoCons on three types of model structures and two application scenarios. Results show that InfoCons effectively extracts interpretable concepts that satisfy both faithfulness and conceptual coherence.

## 2. Background

### 2.1. Point Cloud Models

We focus on a category of point cloud models known as point-based models, which directly handle 3D points *without voxelization or projection*. Point-based models can be formalized as consisting of an encoder and a task-specific head. Encoder $\mathcal{F}$ extracts point features $z \in \mathbb{R}^{D \times N'}$ from input $x \in \mathbb{R}^{(3+C) \times N}$ (3D coordinates with $C$ additional features), and compresses them into a global embedding $z_{\text{global}} \in \mathbb{R}^D$ using symmetric functions that account for the unordered nature of point clouds, typically via maxpooling or a combination of max- and avgpooling. For shape classification, the head outputs a shape prediction $y$. In segmentation tasks, the head typically receives a concatenated point- and global-level feature $z_{\text{concat}} \in \mathbb{R}^{2D \times N'}$ to output point labels (Qi et al., 2017a).

**Information Redundancy in Point Cloud.** Voxel- or projection-based PC models reduce information redundancy at the input level, but this approach often leads to significant information loss, limiting model performance (Liu et al., 2019). Point-based models employ a different strategy. Hierarchical models use sampling operations in each encoder

block, reducing the number of point from $N$ to $N' < N$ layer-by-layer. Non-hierarchical models address redundancy by applying symmetric functions $f : \mathbb{R}^{D \times N'} \to \mathbb{R}^D$ after extracting point features, ensuring that shape-relevant information is preserved.

## 2.2. Information Bottleneck Theory

The Information Bottleneck (IB) theory (Tishby et al., 2000) provides a theoretic perspective on defining what we mean by a 'good' representation. The IB models the role of deep networks as learning a minimal sufficient statistics (maximally compressed) $z = f(x, \theta)$ for predicting the class label $y$. The IB principle can be formalized as a tradeoff between having a precise representation (being informative to $x$) and achieving good predictive power (being informative to $y$):

$$J_{\text{IB}} = \max_{\theta \in \Theta} I(z(\theta), y) - \beta I(x, z(\theta)), \quad \beta > 0,$$

where $I(\cdot, \cdot)$ denotes the mutual information, capturing all non-linear relationships between the two random variables; and $\beta$ controls the degree of restraining input-relevant information. Variational Information Bottleneck (VIB) (Alemi et al., 2016) provides a variational approximation to Information Bottleneck, allowing us to parameterize the IB objectives by training a neural network. See Appendix A.1 for details.

## 2.3. Attribution Methods

**Desiderata of Explanations.** The desiderata of explanation methods can be traced back to LIME (Ribeiro et al., 2016), which aims to provide explanations for classifier predictions in an *interpretable* and *faithful* manner by approximating the behavior of a complex model with a local, interpretable surrogate. Grad-CAM (Selvaraju et al., 2017) introduces a principle for what constitutes a good visual explanation in image classification: explanations should be *class-discriminative* (faithfully localizing objects of interest) and *high-resolution* (capturing fine-grained details that make explanations interpretable).

While this principle is well-suited for image data, where clear boundaries often exist between objects of interest and unimportant background pixels, such boundaries are typically absent in point cloud data, which consists of thousands of unordered 3D points. Concept-level explanations offer a more general formulation of interpretability (Koh et al., 2020). In the image domain, concepts can be human-labeled attributes (Koh et al., 2020) or discovered as selective input patches (Brendel & Bethge, 2019; Zhou et al., 2024).

**VIB for Attribution.** Some works have explored the way of utilizing the VIB objectives to solve the problem of attributing the black-box decisions of deep models (Liu et al., 2024b). Given a pre-trained image classifier, Schulz et al.

(2020) propose to identify the importance of each pixels by optimizing the pixel-wise intensity scale $\lambda$ of a random noise $\epsilon$, which is applied to the latent feature $f(x)$ in an additive manner: $z = \lambda f(x) + (1 - \lambda)\epsilon$, $\epsilon \sim N(\mu_f, \sigma_f^2)$, where $f$ denotes intermediate layers, $f(x) \in \mathbb{R}^{H_f \times W_f \times D_f}$, and a `sigmoid` function constraints $\lambda \in [0, 1]^{H_f \times W_f \times D_f}$. $\lambda$ is then interpolated to the input size and averaged along the channel dimension to obtain a pixel-level attribution map $\Lambda \in \mathbb{R}^{H_I \times W_I}$ (assuming that all channel dimensions are *i.i.d.*). The VIB objectives for attributing (VIB-A) (Schulz et al., 2020) can be formalized as a combination of the classification cross-entropy loss and the information loss:

$$L_{\text{VIB-A}} = - \mathbb{E}_{x \sim p(x), z \sim p(z|f(x))} \left[ \int p(y|z) \log q(y|z) dy \right]$$
$$+ \beta \mathbb{E}_{x \sim p(x)} \underbrace{[D_{KL}(p(z|f(x)) \| q(z))]}_{\mathcal{L}_I}, \quad (1)$$

where the prior is defined as $q(z) = N(\mu_f, \sigma_f^2)$. Intuitively, the VIB-A objectives is optimized to first find the pixels that encode the least information about the label and replace their feature values with random Gaussian noise, resulting in $\Lambda$ having a low saliency score for background pixels.

In our work, we formalize the desiderata of a good critical subset in point cloud attribution: it should be faithful to the model's predictions and conceptually coherent with human priors. To this end, we apply information-theoretic principles to identify critical concepts that satisfy these properties.

## 2.4. Attribution for Point Cloud

**Pooling-based Methods.** Pooling-based methods derive score maps directly from activation maps. For example, Critical Points (CP) (Qi et al., 2017a) identifies a subset of points as critical if they remain active after the final max-pooling layer of the point encoder, and Critical Points++ (CP++) (Levi & Gilboa, 2023) extends CP to continuous measures by applying meanpooling. Many point cloud models also use feature map statistics (averages or maxima) to heuristically explain their proposed models. However, these approaches are primarily based on the encoder of point cloud models and often overlook the influence of the downstream classifier, rendering them insufficient for fully explaining model decisions. Besides, FFAM (Liu et al., 2024a) leverages the non-negative matrix factorization (NMF) of feature maps tailored for attributing 3D object detection results.

**Gradient-based Methods.** PC Saliency Map (PC-SAM) (Zheng et al., 2019) identifies critical points based on the negative gradient of the discriminative loss with respect to points' *spherical coordinates*. Several other works on adversarial attacks also use gradient-based critical points (with C&W loss) as guidance for optimizing point perturbations (Xiang et al., 2019; Huang et al., 2022). These methods tend to produce locally aggregated critical points at spatial

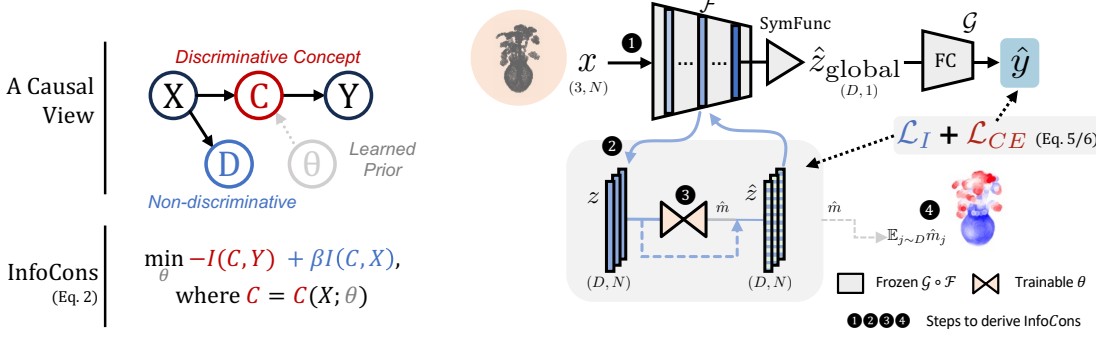

(a) Theoretical Illustration             (b) Framework Overview

*Figure 2.* (a) Theoretical illustration for extracting critical concepts. (b) Overview of our explanation framework to obtain the InfoCons. PC model for shape classification is denoted as $\mathcal{G} \circ \mathcal{F}$. SymFunc stands for symmetric functions (e.g., maxpooling). Attention Bottleneck $\theta$ is trained with end-to-end objectives. Once trained, $\theta$ uses only the intermediate feature to provide explanations.

corners, observed across various point cloud models, even those with significantly different architectures (as shown in Fig. 4). This is caused by the biased prior introduced (we refer to as the shape prior of a *PC centroid*), leading to model-independent explanations (Hong et al., 2023; Feng et al., 2024).

**Black-box Query-based Methods.** LIME3D (Tan & Kotthaus, 2022) extends LIME (for images) to explain point cloud classification by creating a set of queries by flipping the original sample and then fitting a surrogate model to predict point scores. LIME3D uses point-dropping as the flipping operation, considering the discrete nature of point clouds. Similarly, OccAM (Schinagl et al., 2022) proposes a perturbation-based (point-dropping) method to explain object detection results (*e.g.*, a detected car with bounding box), estimating the influence of each point using a sampling-and-forward brute-force approach.

## 3. Methodology

### 3.1. Overview: IB for Critical Points

For deriving a good critical subset that encodes sufficient shape-relevant information (faithfulness) and conforms a semantically meaningful structure (conceptualness), we rewrite the IB objectives as follows:

$$J_{\text{IB-CP}} = \max_{\mathcal{C}} I(\mathcal{C}, y) - \beta I(x, \mathcal{C}), \quad (2)$$

where $\mathcal{C} = m \odot x \subseteq x$ denotes a predictive critical subset of point cloud $x$ by applying a binary mask $m$, $\beta$ controls the tradeoff between being predictive and being less redundant.

Following the variational approach, we use cross-entropy for classification as a lower bound of $I(\mathcal{C}, y)$ (Xie et al., 2020), but estimating $I(\mathcal{C}, x)$ is non-trivial since we *do not have knowledge about the prior* $p(\mathcal{C})$ (Lin et al., 2023). To address this issue, one straightforward solution is to formalize $\mathcal{C}$ as

a discrete selection of points where $m$ is a random variable sampled from a multinomial distribution and the selection probability represents the importance of points:

$$m \sim \text{Mul}(D_r, \text{softmax}\{s(x)_i\}_{i=1}^N),$$
$$m_j \sim \text{Cat}(\text{softmax}\{s(x)_i\}_{i=1}^N), \quad j \in \{1, \cdots, D_r\}, \quad (3)$$

where $D_r$ represents the **r**educed size dimension of $\mathcal{C}$, which is bounded by dimension bottleneck[2]; and $s(x)_i = s(x_i) \in \mathbb{R}$ is the probability of point $i$ being selected as a critical point. In order to derive a soft and continuous score map for $x$, we use a soft relaxation of $m$ denoted as $\hat{m} \in \mathbb{R}^{D_r \times N}$. We leverage a neural network with learnable parameters $\theta$ to calculate the soft mask $\hat{m}$ and apply it to point features $z$:

$$\hat{m} = f(\hat{m}|z(x); \theta), \quad \hat{z} = \hat{m} \odot z(x), \quad (4)$$

where $\odot$ represents element-wise multiplication, and $z$ is the intermediate feature of a given trained PC model by $z(\cdot) = \mathcal{F}^{1:l}(\cdot)$ at $l$-th layer. Noted that $z_{:,i}(x) = z(x_i)$ is the $D$-dim feature of $x_i$, thereby $\hat{m}$ optimized for $z$ is also applicable for $x$.

**Definition 3.1** (Selective Critical Points). With the formulation above, the objectives in Eq. 2 can be formalized as:

$$\max_{\theta} \mathop{\mathbb{E}}_{x \sim p(x)} [\underbrace{\mathbb{E}_{y \sim p(y)} \log q(y|\hat{z})}_{-\mathcal{L}_{CE}} - \beta \underbrace{D_{KL}(\hat{m}||q(\hat{m}))}_{\mathcal{L}_I}], \quad (5)$$

where we replace $q(y|m \odot x)$ by $q(y|\hat{z})$. Considering $I(z(x), x)$ is constant for a given encoder $\mathcal{F}$, we approximate the upper bound of $I(x, \mathcal{C})$ by $D_{KL}(\hat{m}||q(\hat{m}))$, where $q(\hat{m})$ is the prior distribution by setting $q_{:,i}(\hat{m}) = U(0, 1)$ and we use a sigmoid function to constraint $\hat{m} \in \mathbb{R}^{[0,1]}$.

---

[2]Dimension bottleneck proposed by Qi et al. (2017a) refers to the maxpooling layer used to obtain the critical subset where the channel dimension (denoted as $D$) implicitly bounds the size of critical subset $\mathcal{N}_{\mathcal{C}}$.

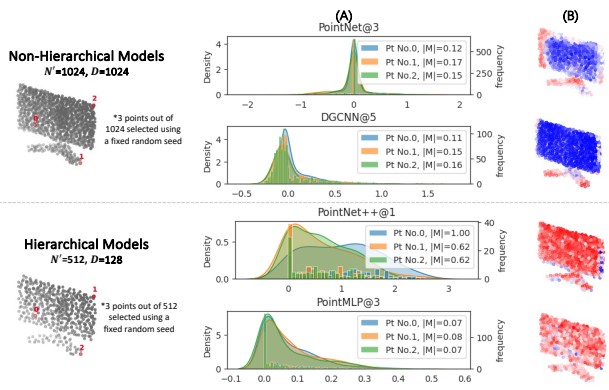

Figure 3. *Feature Analysis* for four models in (A), and derived *Selective Scores* in (B). For PointNet, points with stronger activation receive higher scores (*e.g.,* Point No. 1,2 in (A), which have larger means of activation maps $|M|$). However, in hierarchical models, nearly all points are highlighted (Point No. 0,1,and 2 in (B)). We address this problem by proposing InfoCons in Eq. 6.

Once the $\theta$ is trained, the underlying probabilities $s(x) \in [0,1]^N$ for sampling $m$ can be obtained by calculating the expectation of $\hat{m}$ by $s(x) = \mathbb{E}_{j\sim D_r} \hat{m}_j$, as shown in Fig. 2-(b).

**Feature Analysis.** Selective CP defined in Eq. 5 is effective for non-hierarchical, localized PC models like PointNet (Qi et al., 2017a). PointNet enables these objectives to effectively balance the CE loss and information loss through point-wise masking, largely due to the high sparsity of its point features. As shown in Fig. 3, the features from Point-Net in (A) are significantly sparser than those from Point-Net++. Additionally, the selective scores for PointNet for three selected points (Point No.0,1,2) align with their absolute mean feature values $|M|$ (*i.e.,* $1 > 2 > 0$). In contrast, in PointNet++, higher $|M|$ (and thus higher selective scores) do not correspond to the points that should be critical (by observing $0 > 1 = 2$). The feature analysis explains why CP++ (meanpool-based) fails with PointNet++ (in Fig. 4).

### 3.2. Deep InfoCons

Recent PC models usually enhance the feature extraction process by incorporating with feature grouping operations (e.g., DGCNN), set abstraction (e.g., PointNet++), or deeper stacked layers with residual connections (e.g., PointMLP), where the information flow from points to intermediate features becomes highly non-linear, resulting in inaccuracy for approximating $I(\mathcal{C}, \cdot)$. More specifically, the score map $\hat{m}$ is ineffective as most of points are considered with equal importance, as shown in Fig. 3. This is mainly due to the entanglement of point features with their neighbours when we try to measure the importance of individual points. We introduce InfoCons as follows to address this problem:

**Definition 3.2** (InfoCons). Informative scores with informa-

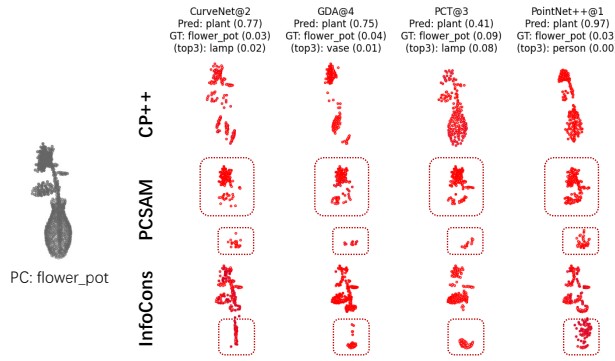

Figure 4. InfoCons-based critical subsets (200 pts) for four PC models (covering three distinct structure types) are compared with meanpool-based CP++ and gradient-based PCSAM. PCSAM tends to extract similar and spatially aggregated subsets for all models, while InfoCons identifies more interpretable critical subsets that are faithful to model behavior (color distinction not required).

tive concepts for a given PC model can derived as follows:

$$\max_\theta \mathbb{E}_{x\sim p(x)}[\mathbb{E}_{y\sim p(y)} \log q(y|\hat{z}) - \beta D_{KL}(\hat{z}||q(\hat{z}))],$$
$$\hat{z} = \hat{m} \odot z(x) + \text{sg}(1 - \hat{m}) \odot \epsilon, \quad (6)$$

where $\odot$ denotes element-wise multiplication, $\text{sg}(\cdot)$ denotes stop-gradient operation. $\epsilon_i \sim N(\mu_z, \sigma_z^2)$ is sampled for each point $i$. Gaussian prior $q(\hat{z}) = N(\mu_z, \sigma_z^2)$ is parameterized with the $D$-dimensional mean and variance of point feature $z = \mathcal{F}^{1:l}(x)$. We reused the neural network to formalize $\hat{z}$ by $\hat{z} = f(\hat{z}|z, \theta)$.

As defined in Def. 3.2, to address the problem of entanglement, we try to decouple the information from neighbour points and the anchor point, by introducing a Gaussian noise denoted as $\epsilon$. We aim to recover a coarse feature about neighbors so that only the information about the anchor point is reduced. More specifically, when we need to rank $x_i$ as unimportant ($\hat{m}_i \to 0$) (for the reason that either it is redundant or less shape-informative), and the feature of $i$-th point has multiple information sources from $k$ neighbours together with the point $x_i$ itself, then we try to add a random variable sampled from a specific distribution determined by $z$, avoiding significant degradation on model performance.

**Comparison with PC Saliency Map (Zheng et al., 2019).** PC Saliency Map is one of the most representative explaining methods for PC data, which can be formalized as: $s_i(x) = -\partial\mathcal{L}_{CE}/\partial r_i \cdot r_i^{1+\alpha}$ ($r_i$ denotes the distance of $x_i$ to the spherical core and $\alpha$ is a scale factor for varied spatial sizes of PC). This score introduces prior that the points lying in the surface of objects tend to be more salient, resulting in redundant critical points clustered in spatial corners, as shown in Fig. 4-(PCSAM).

**Attention Bottleneck.** The intermediate features of point encoders are high-correlated and entangled, particularly in

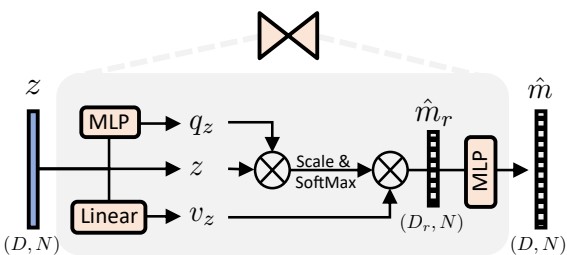

*Figure 5.* The data flow of the bottleneck network $f(\cdot|z(x);\theta)$. The input $z$ is the intermediate feature from $\mathcal{F}$, and the output $\hat{m}$ holds the same dimension as $z$.

the hierarchical models and self-attention based models. We introduce a non-linear attention block to learn the unbiased priors based on point-level features, denoted as $f(\hat{m}|z;\theta)$, as shown in Fig. 5.

The input of attention bottleneck is the intermediate feature of point encoder, denoted as $z$. For hierarchical models (*e.g.*, PointNet and DGCNN), $z$ holds the same size dimension as input $x$ ($N' = N$ and we use $N = 1024$). However, for the hierarchical models, points are down-sampled or pre-processed to form 3D patches, the size dimensions may vary. We apply the channel-wise attention to fit with this property. Formally, the input feature is first transformed as $q_z = W_q^T z, v_z = \sigma(W_v^T z)$, where $W_q^T \in \mathbb{R}^{D \times D_r}, W_v^T \in \mathbb{R}^{D \times D}$ and $\sigma$ represents an `ELU` activation function. Then we calculate the output:

$$\text{Att}(q_z, z, v_z) = \text{softmax}(q_z^T z / \sqrt{D}) \cdot v_z. \qquad (7)$$

The output is finally expanded to the dimension $D$ through MLPs with a `sigmoid` activation to constraint $\hat{m} \in (0, 1)^{D \times N'}$. In order to visualize the score map in the case of $N' < N$, we further propagate $\hat{m}$ to $N$ using spatial interpolation (Qi et al., 2017b) (weighted by L2 distance between $N'$ anchor points and $N$ original points).

## 4. Experiments

**Evaluation Protocol.** The example point clouds are selected from the synthetic dataset ModelNet40 (Wu et al., 2015) and two real-world datasets: ScanObjectNN for shape classification (the hardest variant PB_T50_RS (Uy et al., 2019)) and KITTI for objection detection (Geiger et al., 2012). Our evaluations are mainly conducted on ModelNet40, which contains $9,843$ training samples and $2,468$ testing samples. We uniformly sample $1,024$ points from the surface of CAD models and only 3D coordinates are used (*i.e.*, $x \in \mathbb{R}^{3 \times 1,024}$). We compare InfoCons with four baselines: Critical Points (Qi et al., 2017a), PC Saliency Map (Zheng et al., 2019), Critical Points++ (Levi & Gilboa, 2023) and black-box query-based LIME3D (Tan & Kotthaus, 2022). One of the quantitative metrics used is overall accuracy (OA) under point-drop attacks, calculated as:

Acc $= \frac{1}{\mathcal{N}_\mathcal{X}} \sum_i \mathbb{I}(\hat{y}_i = y_i)$, where $\mathcal{N}_\mathcal{X}$ denotes for the size of attack dataset (we use $\mathcal{N}_\mathcal{X} = 2,468$). We conducted comparison experiments on eight PC models: PointNet, CurveNet, GDA, PointMLP, DGCNN, Maskpoint, and PCT (Muzahid et al., 2020; Xu et al., 2021; Ma et al., 2022; Qi et al., 2017b; Wang et al., 2019; Liu et al., 2022; Guo et al., 2021). *Model descriptions* and *training details* are provided in Appendix C. The source code is available at `https://github.com/llfffff/infocons-pc`.

### 4.1. Qualitative Comparison

We compare InfoCons with four baselines for PC explanation in Fig. 6 qualitatively. Critical Points++ (A0, A1), InfoCons (D0, D1), and Critical Points (E) are derived with a single forward pass. PCSAM (C0, C1) is implemented by iteratively dropping points over 20 steps (10 points per step), and LIME3D (B0, B1) performs 1,000 queries using point-dropped point clouds.

- In (A0-D0), we demonstrate the soft score maps on four PCs for DGCNN. The PCs shown are selected from ModelNet40 test split, where the model gives incorrect predictions (probabilities of the ground truths and predictions are shown on the far left).

- In (E, A1-D1), we visualize the derived *critical subset* (200 out of 1024 points) from the score maps, except for CP, where the size of the critical subset is determined adaptively by the maxpooling layer.

- Compared to the baselines, our InfoCons method successfully attributes the incorrect predictions to the corresponding concepts with greater informativeness and reduced redundancy. Specifically, the mislabeled '*plant*' in (R1) arises from missing the '*pot*'; and the confused '*flower_pot*' in (R2) can be attributed to the model's attention on the '*pot*'. For samples (R3) and (R4), InfoCons (D0 and D1) outperforms baselines by exhibiting less redundancy of similar points and providing richer shape-relevant information.

In Fig. 11 of Appendix B, we visualize information redundancy of a given PC by applying K-Means clustering to the InfoCons-based score map, forming a *Critical-Subset Hierarchy*. In Fig. 13, we demonstrate the *Dynamic Critical Subset* by iteratively dropping points and *re*-constructing score maps, providing a fair qualitative comparison with PCSAM.

### 4.2. Quantitative Comparison

**Effectiveness of InfoCons.** Following Zheng et al., we conduct a point-drop attack to evaluate the effectiveness of InfoCons by measuring the accuracy changes of **(i)** dropping out the most critical points (MCD) and **(ii)** dropping

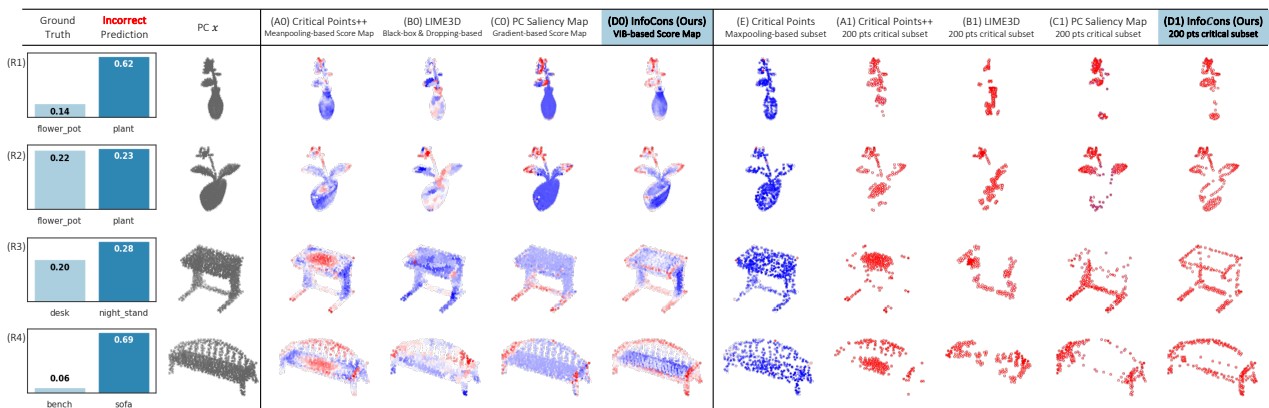

Figure 6. Qualitative comparisons of InfoCons with four baselines (CP, CP++, PCSAM, LIME3D) include both score maps (best viewed in color) and critical subsets. The PCs are selected from ModelNet40, where DGCNN gives incorrect predictions.

out the least critical points (LCD). We report the instance accuracy for 1pass score maps and multi-iteration (from 5 to 20) dynamic score maps in Fig. 7. We iteratively drop 10 LC/MC points each iteration, thus *InfoCons (20it)-MCD* denotes score maps for 824 points (200 most critical points dropped). The accuracy gap between MCD and LCD indicates InfoCons captures the shape-informative points by assigning them higher scores.

**Comparison with baselines.** Based on point-drop attacks, we further compare the proposed method with three baselines with soft score map (CP++, PCSAM and LIME3D) in Fig. 7-(B,C). Since PCSM is a gradient-based method that utilize the ground truth label to calculate the loss, our Info-Cons may not outperform PCSAM when dropping only a small critical subset. However, our method InfoCons (1pass) achieves better performance when remove a larger subset, indicating the information redundancy of critical concepts can be appropriately handled by InfoCons. Besides, LIME3D achieves the most substantial drop-attack by significantly degrading the accuracy to 45.22% and InfoCons (20iter) ranks second with 63.70% (at 500-pts-dropping point). However, LIME3D incurs a heavy computational cost (about 4.5s/PC shown in Tab. 3) limiting its applications. The interpretability of LIME3D also lacks, as it is challenging to interpret the semantics of the derived critical subsets compared to white-box methods. Full evaluations on eight models are in Fig. 18 and Fig. 19 of Appendix D.2.

### 4.3. Extensive Evaluations

We integrate InfoCons into two applications by enhancing their salient-region search algorithm, and successfully achieving improved performance (Fig. 8). We also extend InfoCons on two challenging real-world benchmarks ScanObjectNN and KITTI (Fig. 10).

**InfoCons for Data Augmentation.** We replace the gradient-

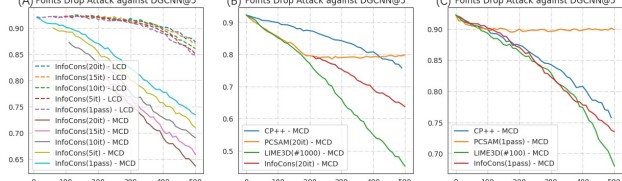

Figure 7. Accuracy under point-drop attacks against DGCNN. We compare InfoCons with CP++, PCSAM, and LIME3D. In (A), the effectiveness of InfoCons is demonstrated by the accuracy gap between LCD and MCD. In (B) and (C), we present comparisons in the *most-powerful setting* and *fair-running-time setting*, with efficiency details provided in Table 3.

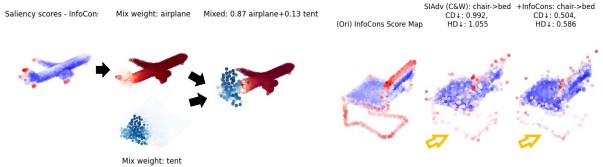

Figure 8. InfoCons-based mixup (left) and adversarial attack (right). InfoCons-based score map can also explain the effectiveness of adversarial examples. Better viewed in color.

based saliency score map of SageMix (Lee et al., 2022) with InfoCons, then following SageMix's dynamic data augmentation to generate continuous mixed PC. We re-implement SageMix and report the results in Tab. 1. We use a larger batch size and fewer training epochs (96, 300) compared to the original paper *(32, 500)*. Under the same training hyper-parameters, we achieve improved classification performance by integrating InfoCons into the mixup pipeline, primarily because InfoCons mitigates the gradient saturation effect through KL divergence regularization.

**InfoCons for Adversarial Attack.** We integrate the proposed method into SI-Adv (Huang et al., 2022) to generate effective and imperceptible adversarial perturbations. SI-Adv is a sensitivity-guided attack, and we enhance its

*Table 1.* Quantitative evaluations of InfoCons for mixup. Our re-implemented results are shown in black, with SageMix's original results displayed in *(grey)*.

| Model | Aug Type | Test OA↑ (%) |
|-------|----------|--------------|
| DGCNN | Base | 92.30 *(92.9)* |
| DGCNN | SageMix | 92.79 *(93.6)* |
| | +InfoCons | **93.19** (+0.4) |

*Table 2.* Quantitative evaluations of InfoCons for adversarial attacks. Enhanced with InfoCons, we can achieve higher ASR and lower average CD/HD.

| DGCNN (on ModelNet40 testset) | | | |
|-------------------------------|-------|-----------|------|
| | SI-Adv | +InfoCons | Δ |
| ASR↑ (%) | 99.76 | **99.80** | *(+0.04)* |
| $\overline{CD}$↓ $(10^{-1})$ | 5.58 | **5.47** | *(−0.11)* |
| $\overline{HD}$↓ $(10^{-1})$ | 6.70 | **6.55** | *(−0.15)* |

sensitive-region search by rescaling the gradient from C&W loss using InfoCons' score map. For fair comparison, we use the same step size and normalize the resulting score maps for both SI-Adv (+Sensitive scores, original) and SI-Adv (+InfoCons). We use a step size of 0.007 and 80 steps in total. As in Fig. 8 (right) and in Tab. 2, InfoCons provides more effective salient regions that help optimize adversarial perturbations. The score maps derived from InfoCons can also explain the effectiveness of adversarial examples (e.g., changes in the scores around the leg of a perturbed 'chair' contribute to the success attack that causes the model to misclassify it as a 'bed').

*Table 3.* Efficiency comparisons of InfoCons with four baselines.

| Method | OA(%)↓ | $ET_t$ | $ET_e$ (s)↓ | #Param |
|--------|--------|--------|-------------|--------|
| CP++ | 75.08 | 1**F** | **0.01** | 0 |
| PCSM(1pass) | 89.87 | 1(**F+B**) | 0.05 | 0 |
| PCSM(20it) | 79.86 | 20(**F+B**) | 0.85 | 0 |
| LIME3D($10^2$) | 67.99 | 100**F** | 0.54 | 1K |
| LIME3D($10^3$) | **45.22** | 1000**F** | 4.54 | 1K |
| InfoCons(1pass) | 73.50 | 1**F** | **0.01** | 2.4M |
| InfoCons(10it) | 69.08 | 10**F** | 0.17 | 2.4M |
| InfoCons(20it) | 63.70 | 20**F** | 0.29 | 2.4M |

1. **F**: one forward-pass time, **B**: one backward-pass time.
2. Empirical explaining time ($ET_e$) is measured using the `%timeit` magic command in Jupyter Notebook.

### 4.4. Efficiency and Parameter Study

**Efficiency Study.** In Tab. 3, we report (1) test overall accuracy (OA) after dropping the 500 most critical points, where a lower OA indicates a more effective explanation following Zheng et al. 2019; Tan & Kotthaus 2022; (2) theoretical explanation time for each point cloud *($ET_t$)* and empirical explanation time *($ET_e$)*; and (3) the size of the explainers. InfoCons and CP++ require only a single forward pass, making them the most time-efficient methods, which is an important advantage for applications like dynamic data aug-

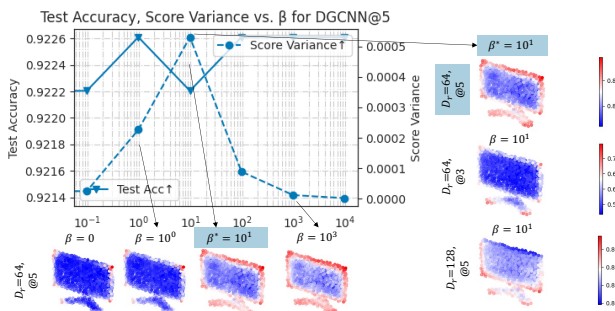

*Figure 9.* Impacts of $\beta$, $D_r$ and intermediate layer $l$ for DGCNN@$l$ on ModelNet40.

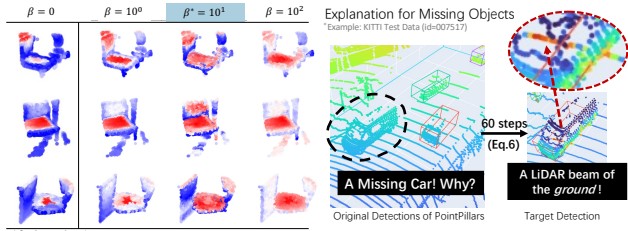

*Figure 10.* Impacts of $\beta$ for DGCNN@5 on ScanObjectNN (left) and the explanation for missing objects of object detector PointPillars@1 (right, better viewed in color). The implementation details can be found in Appendix C.2.

mentation and adversarial attacks. The results are derived for DGCNN on ModelNet40 testset. LIME3D achieves the best drop attack performance by querying $10^3$ times and *InfoCons (20iter)* ranks the second. However, there is a limitation in using the drop rate of OA as an effectiveness metric, as a larger drop may result from removing points in certain spatial locations that push the point cloud off the data manifold, rather than reflecting true faithfulness in terms of contributions to the predicted labels.

**Parameter Study.** The proposed framework is occupied with several hyper-parameters to choose, we demonstrate three key hyperparameters of our InfoCons in Fig. 9:

- **The weighting coefficient** $\beta$ applied to the information loss, which controls the level of retaining point-wise positional information. We utlize the variance of scores as a metrics to measure the quality of the score map, and consider the optimal choice of $\beta$ together with the test set accuracy. We empirically observe that there exists an optimal point for the choice of $\beta$ that reaches relatively high accuracy with high score variance.

- **The reduction dimension** $D_r$ of attention bottleneck, which impacts the prior distribution. A smaller $D_r$ will be more parameter-efficient. We observe that different $D_r$ have varying optimal $\beta$ values but do not significantly alter the optimal score map distribution, and we set $D_r = 64$ as the default value.

- **The intermediate layer** $l$ impacts the target feature map $z = \mathcal{F}^{1:l}$ that InfoCons uses to identify informative concepts. A basic principle for choosing $l$ is that the layer should be as deeper as possible to be shape-informative, while ensuring the size dimension of $z$ is not too small ($N' > 256$) to reduce the entanglement of neighbour features. This is crucial for the visualization quality of the score maps (measured by the variance). Following this, we manually determine the intermediate layer for eight models, listed in Appendix C.4.

## 5. Conclusion

In our work, we propose a PC explanation framework called InfoCons, which attributes model predictions to critical concepts. We provide an in-depth analysis of the limitations of existing attribution methods and design an IB-based objective to derive a faithful and interpretable critical score map. This enables humans to identify the concepts that the model correctly or unexpectedly focuses on. We demonstrate the effectiveness of the proposed methods both qualitatively and quantitatively. InfoCons is also scalable across multiple structures of point cloud models (eight models in total) and real-world datasets. Furthermore, the application of InfoCons in saliency-based mixup and adversarial attacks further demonstrates its effectiveness.

The main limitation of our method is the computational load required for optimizing the attention bottleneck module: Some PC models may have low computational efficiency, and the selection of some hyperparameters such as $\beta$ and the intermediate layer $l$ is critical. We demonstrate some failure cases in Appendix D.1. Future direction could include evaluating and mitigating the bias in PC pre-trained models and improving their generalization ability under domain shifts.

## Acknowledgement

We are thankful to the shepherd and reviewers for their careful assessment and valuable suggestions, which have helped us improve this paper. This work was supported in part by the National Natural Science Foundation of China (62472096, 62172104, 62172105, 62102093, 62102091, 62302101, 62202106). Min Yang is a faculty of the Shanghai Institute of Intelligent Electronics & Systems and Engineering Research Center of Cyber Security Auditing and Monitoring, Ministry of Education, China.

## Impact Statement

This paper aims at improving the faithfulness and interpretability of attribution methods for PC classification models. The method is applicable to safety-critical scenarios including autonomous vehicles, where our approach can help the human inspects to better understand the root cause of accidents or unexpected decisions of LiDAR-based deep learning models in the vehicles.

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

# A. Theoretical Details

## A.1. Deep Variational Information Bottleneck

The Information Bottleneck (IB) provides the principle of what is a good representation in terms of a fundamental tradeoff between being predictive to labels and keeping maximal redundant information of input data (Alemi et al., 2016). Let $x$ denotes inputs and $y$ denotes the corresponding target labels, we can regard the output of intermediate layers of a network as a stochastic encoding $z$ *w.r.t* the input $x$. For correctly classify the input $x$ as $y$, our primary goal is to learn a encoding function that maximize the information about $y$ in $z$:

$$\max_z I(z, y),$$

which upper bound given by the dataset $I(x, y)$ is independent to the network and the encoding $z$. To prevent the trivial solution ($z = x$), the IB principle apply a constraint $I_c$ to $I(z, x)$, and we can use the Lagrange multiplier $\beta$ to get the objective functions:

$$\max_z I(z, y) \text{ s.t.} I(x, z) <= I_c$$
$$\max_z I(z, y) - \beta I(x, z),$$

where the hyper-parameter $\beta > 0$ controls the degree of restraining information about the input. The first item can be expressed as follows:

$$I(z, y) = \int p(x, y) \log \frac{p(y, z)}{p(y)p(z)} dydz$$
$$= \int p(y, z) \log \frac{p(y|z)}{p(y)} dydz$$

For the intractable term of likelihood $p(y|x)$, we can introduce $q(y|z)$ to approximate $p(y|z)$ by a classifier $g : Z \to Y$. With the nonnegativity of the Kullbach-Leibler divergence:

$$D_{KL}[p(y|z)||q(y|z)] \geq 0 \Rightarrow \int p(y, z) \log p(y|z) dy$$
$$\geq \int p(y, z) \log q(y|z) dy,$$

then we can replace $p(y|z)$ with $q(y|z)$ and derive the lower bound of $I(z, y)$:

$$I(z, y)$$
$$= \int p(y, z) \log \frac{p(y|z)}{p(y)} dydz$$
$$= \int p(y, z) \log p(y|z) dydz - \int p(y, z) \log p(y) dydz$$
$$= \int p(y, z) \log p(y|z) dydz - \int p(y) \log p(y) dy$$
$$= \int p(y, z) \log p(y|z) dydz + H(y)$$
$$\geq \int p(y, z) \log q(y|z) dydz \tag{8}$$

The second term $I(z, x)$ can also be expanded by definition:

$$I(z, x) = \int p(z, x) \log \frac{p(z, x)}{p(x)p(z)} dzdx$$
$$= \int p(z, x) \log \frac{p(z|x)}{p(z)} dzdx$$

Introducing a prior $q(z)$ with learnable parameters to approximate the unknown true distribution of latent representations $p(z)$, we can derive the variational upper bound of $I(z, x)$:

$$D_{KL}[p(z)||q(z)] \geq 0 \Rightarrow$$
$$\int p(z) \log p(z) dz \geq \int p(z) \log q(z) dz$$
$$I(z, x) \leq \int p(z, x) \log \frac{p(z|x)}{q(z)} dzdx$$
$$= \int p(z|x)p(x) \log \frac{p(z|x)}{q(z)} dzdx \tag{9}$$
$$= \mathbb{E}_{x \sim p(x)} [D_{KL}(p(z|x)||q(z))]$$

Combining the lower bound of $I(z, y)$ (Eq. 8) and the upper bound of $I(z, x)$ (Eq. 9), the objective of Information Bottleneck can be reformulated as :

$$\int p(y, z) \log q(y|z) dydz - \beta \underbrace{\mathbb{E}_{x \sim p(x)} [D_{KL}(p(z|x)||q(z))]}_{\mathcal{L}_I}.$$
$$\tag{10}$$

Note that $p(y, z) = \mathbb{E}_{x \sim p(x)} p(y, z|x) = \mathbb{E}_{x \sim p(x)} p(y|z)p(z|x)p(x)$, then the first item can be written as (Schulz et al., 2020):

$$\int p(y|z)p(z|x)p(x) \log q(y|z) dydzdx \tag{11}$$
$$= \mathbb{E}_{x \sim p(x), z \sim p(z|x)} [\int p(y|z) \log q(y|z) dy] \tag{12}$$
$$= \mathbb{E}_{x \sim p(x), z \sim p(z|x)} [-\mathcal{L}_{CE}] \tag{13}$$

### A.2. VIB for Critical Points

Following Eq. 8 for deriving a lower bound of $I(z, y)$ and Eq. 9 for an upper bound for $I(z, x)$, we can formalize the objectives $I(\mathcal{C}, y) - \beta I(x, \mathcal{C})$ defined in Eq. 2 to a variational form:

$$\int p(y, \mathcal{C}) \log q(y|\mathcal{C}) dy dx -$$
$$\beta \int p(\mathcal{C}|x) p(x) \log \frac{p(\mathcal{C}|x)}{q(\mathcal{C})} d\mathcal{C} dx, \quad (14)$$

where we parameterize $\mathcal{C}$ as $m \odot x$ and further approximate it as $\hat{m} \odot z(x)$. $\hat{m}$ is derived from a neural network $f(\hat{m}|z(x); \theta)$ with learnable parameter $\theta$, as proposed in Eq. 5. We omit the intermediate steps $z(x) = \mathcal{F}^{1:l}(x)$, $\hat{m} = f(\hat{m}|z(x); \theta)$ and denote $\hat{m} \odot z(x)$ as $\hat{z} = f(\hat{z}|x; \theta)$ for simplicity. Therefore, objectives in Eq. 14 can be rewritten as:

$$\int p(y, \hat{z}) \log q(y|\hat{z}) dy dx -$$
$$\beta \int p(\hat{z}|x) p(x) \log \frac{p(\hat{z}|x)}{q(\hat{z})} d\hat{z} dx, \quad (15)$$

where the first term can be represented by $-\mathcal{L}_{CE}$. The second term can be reformulated as a KL divergence and further simplified as:

$$\mathbb{E}_x[D_{KL}(p(\hat{z}|x)||q(\hat{z}))] = \mathbb{E}_x[D_{KL}(\hat{z}||q(\hat{z}))] \quad (16)$$

considering $\hat{z} = f(\hat{z}|x; \theta)$. Moreover, since $z(x)$ is determined by the trained encoder $\mathcal{F}$, we consider the upper bound based on $\hat{m}$ in Eq. 5 as follows:

$$I(\mathcal{C}, x) \leq \mathbb{E}_x[D_{KL}(\hat{m}||q(\hat{m}))]. \quad (17)$$

For the calculation of KL divergence, we use the Monte Carlo algorithm to approximate the value at the **point level** (*i.e.* we assume that each point feature is *i.i.d.*, so the large number of points in a single point cloud, together with a relatively small batch size, provides a good approximation performance).

## B. Quantitative Comparisons

**Critical Subset Hierarchy.** Due to information redundancy, a point cloud can be split into different levels of subsets, with each level holding a certain geometric structure. In Fig. 11 and 12, we compare the *Critical Subset Hierarchy* of InfoCons with Critical Points(CP). In CP, the four-level hierarchical subsets are constructed by iteratively dropping critical points over four iterations. For InfoCons, we construct the hierarchical subsets by applying K-Means

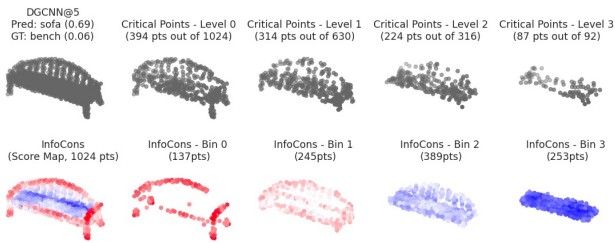

*Figure 11.* Decompose point cloud redundancy by visualizing the *Critical Points Hierarchy*.

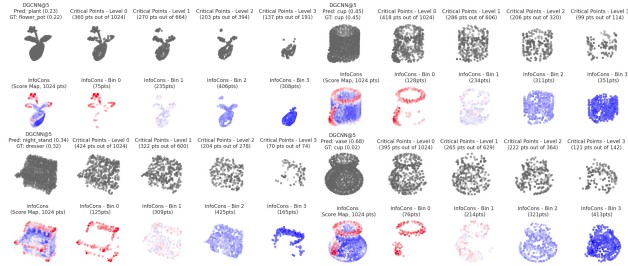

*Figure 12.* Additional samples for *Critical Points Hierarchy*.

clustering ($K = 4$). We observe that the hierarchy of InfoCons suggests the most critical concept is *the contour of a 'sofa'* (as the model incorrectly predicts), and the least critical concept is *the seat* (a common attribute between 'sofa' and 'bench').

**Dynamic Critical Subset.** The dynamic process of iteratively dropping points and constructing gradients to derive score maps is vital for PC Saliency Map (PCSAM) (Zheng et al., 2019). The number of iterations should be large enough to fit the size of the required critical subsets (*e.g.,* 20 iterations for a 200-point critical subset), as demonstrated in Fig. 13. Different from PCSM, InfoCons can produce an effective score map with one-pass score map, and additional iterations (20-iteration) do not significantly change the distribution of the most critical concepts. Therefore InfoCons is more faithful as the PC is not modified during explanation. (Examples for eight models can be found in Fig. 15 and more examples are shown in Fig. 16).

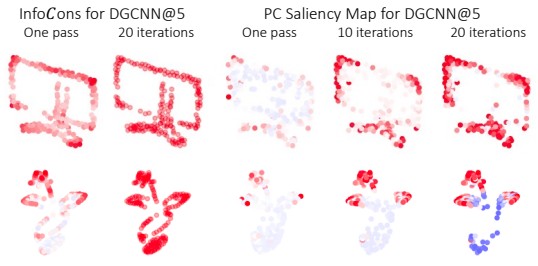

*Figure 13.* Dynamic critical subset (200 points) with one-pass and multi-iteration score maps, compared with PCSAM.

## C. Experimental Details

### C.1. Training Details

For point cloud models involved in our experiments, we follow their official training pipeline[3][4][5][6][7] or open-source re-implementation in Pytorch[8][9][10]. Our experiments of training InfoCons were conducted using an NVIDIA GeForce RTX 2080Ti (11G) GPU, using Python 3.11 and PyTorch 2.1, with CUDA 12.2. For each model, we use a batch size of $k = 8$, and training DGCNN for 50 epochs takes about one hour.

### C.2. InfoCons for Object Detection

We extend the InfoCons framework to explain challenging 3D object detection tasks. The results demonstrate that InfoCons is also effective for object detection. We conduct the study on PointPillars (Lang et al., 2019), and we use the pretrained weights from OpenPCDet[11] and the detection code from the official codebase from OccAM[12].

Given a 3D object detector with point cloud encoder (*e.g.*, PointPillars which contains a Pillar Feature Net) and a PC from KITTI test split, we can derive a score map for explaining missing objects as follows:

- Firstly, we focus on how to explain the missing cases that may cause serious consequences. For example, a missing front-near car (as shown in Fig. 10). Since the KITTI test samples lack ground truth labels, we randomly select data, obtain detection results from the detector, and manually identify the target missing objects.

- For the objectives in Eq.6, we replace $\mathcal{L}_{CE}$ with the detection score of target objects as the surrogate objective for $I(C; y)$. The information loss remains the same as in classification (KL divergence). Specifically, we use the negative of the similarity loss defined in OccAM(Schinagl et al., 2022), as we are explaining *the missing object* rather than the detected one.

- Unlike InfoCons for classification, where the explana-

---

[3] https://github.com/mutianxu/GDANet
[4] https://github.com/Yochengliu/Relation-Shape-CNN
[5] https://github.com/ma-xu/pointMLP-pytorch
[6] https://github.com/tiangexiang/CurveNet
[7] https://github.com/WisconsinAIVision/MaskPoint
[8] https://github.com/yanx27/Pointnet_Pointnet2_pytorch
[9] https://github.com/antao97/dgcnn.pytorch
[10] https://github.com/Strawberry-Eat-Mango/PCT_Pytorch
[11] https://github.com/open-mmlab/OpenPCDet
[12] https://github.com/dschinagl/occam

tion module, AttentionBottleneck, is trained on the entire training dataset, we find it costly to train our explainer on KITTI. Therefore, we directly optimize the explainer on a single-sample (test scene PC) for 60 steps.

As shown in Fig. 10, the score map particularly highlights a short segment of the LiDAR beam located directly beneath the car. Our method reveals that this segment of the ground causes the model to fail in detecting the car.

### C.3. Implementation Details

**Spatial Interpolation.** Spatial interpolation is widely used in hierarchical point models for tasks such as point cloud segmentation. We implement our score map interpolation from a scored subset $(x', m')$ with points set size $N'$ (a sub-sampled result) to the input score map $(x, m)$ with points set size $N$ (the size of input) following KNN-based feature propagation (Qi et al., 2017b). Here, we remove the non-linear learnable transformations for direct attribution, which can be formalized as follows:

$$m_i = \frac{\sum_{j=1}^{k} w_j^{(i)} m_j'^{(i)}}{\sum_{j=1}^{k} w_j^{(i)}}, \text{where } k = 3,$$

$$(w^{(i)}, m'^{(i)}) = \text{index}[(d(x_i, x')^{-1}, m'), \text{sort}[d(x_i, x')]].$$

**Reparameterization Tricks.** The sampling result $m$ from a multinomial distribution is a set of discrete one-hot vectors in our reformulation in Eq. 2 and the process of sampling is undifferentiable. For differentiable optimization, we use Gumbel reparameterization tricks as follows:

$$\hat{m} = \hat{f}(m|z; \theta) = \text{gumbel-softmax}\{f(z_i; \theta)\}_{i=1}^{N}$$

$$= \mathbb{E}_k \text{softmax}\{\frac{g_k + \log f(Z_i, \theta)}{\tau}\}_{i=1}^{N}, \quad (18)$$

where $g_k = -\log(-\log e_k), e_k \sim \mathbf{U}(0, 1)$ is the Gumbel noise and $\tau$ is the temperature. We set $\tau = 0.7$ (larger for a smoother distribution) and sample the noise for $k = 32$ times to calculate the expectation.

### C.4. Point Cloud Models Details

**Model Architectures.** We provide detailed information about the architectures of the point cloud models, as well as the intermediate layers where we apply the Attention Bottleneck in Tab. 4.

**List of available layers in PC models encoder** We provide detailed information about the official architectures of the point cloud models involved in our work in Tab. 5, 6, 7. Additionally, we specify the intermediate layers of these models that we targeted.

*Table 4.* Model descriptions with point feature information.

| PC Model @layer $l$ | Size Dim. $N'$ | Structure of Model | Sparsity/% $\sum \mathbb{I}(z = 0)/(D' \times N')$ | Channel Dim. $D'$ |
|---|---|---|---|---|
| PointNet@3 (Qi et al., 2017a) | 1024 | N-H | 62.30 | 1024 |
| CurveNet@2 (Muzahid et al., 2020) | 1024 | H | 60.19 | 128 |
| GDA@4 (Xu et al., 2021) | 1024 | N-H | 53.94 | 512 |
| PointMLP@4 (Ma et al., 2022) | 64 | H | 24.24 | 1024 |
| PointNet++@1 (Qi et al., 2017b) | 512 | H | 2.31 | 128 |
| DGCNN@5 (Wang et al., 2019) | 1024 | N-H | 0.00 | 1024 |
| MaskPoint@10 (Liu et al., 2022) | 64 | SA | 0.00 | 384 |
| PCT@4 (Guo et al., 2021) | 256 | SA, H | 0.00 | 1024 |

*Table 5.* Non-hierarchical Structure

| | @layer | #points | #channels | |
|---|---|---|---|---|
| PointNet (Qi et al., 2017a) | encoder.2 | 1024 | 128 | |
| | encoder.3 | 1024 | 1024 | ✓ |
| | maxpooling | / | 1024 | |
| DGCNN (Wang et al., 2019) | encoder.3 | 1024 | 128 | |
| | encoder.4 | 1024 | 256 | |
| | encoder.5 | 1024 | 1024 | ✓ |
| | maxpooling | / | 1024 | |
| | avgpooling | / | 1024 | |
| GDA (Xu et al., 2021) | encoder.2 | 1024 | 64 | |
| | encoder.3 | 1024 | 128 | |
| | encoder.4 | 1024 | 512 | ✓ |
| | maxpooling | / | 512 | |
| | avgpooling | / | 512 | |

*Table 6.* Hierarchical Structure

| | @layer | #points | #channels | |
|---|---|---|---|---|
| PointNet++ (Qi et al., 2017b) | encoder.1 | 512 | 128 | ✓ |
| | encoder.2 | 128 | 256 | |
| | encoder.3 | 1 | 1024 | |
| PointMLP (Ma et al., 2022) | encoder.2 | 256 | 256 | |
| | encoder.3 | 128 | 512 | |
| | encoder.4 | 64 | 1024 | ✓ |
| | maxpooling | / | 1024 | |
| CurveNet (Muzahid et al., 2020) | encoder.2 | 1024 | 128 | ✓ |
| | encoder.3 | 256 | 256 | |
| | encoder.4 | 64 | 512 | |
| | encoder.5 | 64 | 1024 | |
| | maxpooling | / | 1024 | |
| | avgpooling | / | 1024 | |

*Table 7.* Multi-head Self-attention-based

|  | @layer | #points | #channels |  |
|---|---|---|---|---|
| MaskPoint (Liu et al., 2022) | sa.6 | 64 | 384 |  |
|  | sa.8 | 64 | 384 |  |
|  | sa.10 | 64 | 384 | ✓ |
|  | maxpooling | / | 384 |  |
|  | <cls> | / | 384 |  |
| PCT (Guo et al., 2021) | encoder.2 | 256 | 256 |  |
|  | encoder.3 | 256 | 1024 |  |
|  | encoder.4 | 256 | 1024 | ✓ |
|  | maxpooling | / | 1024 |  |

# D. Additional Results

## D.1. Failure Cases

We demonstrate the failure cases for PCT in Fig. 14, where appropriate layer $l$ is critical and the changes of $D_r$ (rescale by $\alpha$) and $\beta$ is unable to derive a critical concept which conforming a meaningful structure.

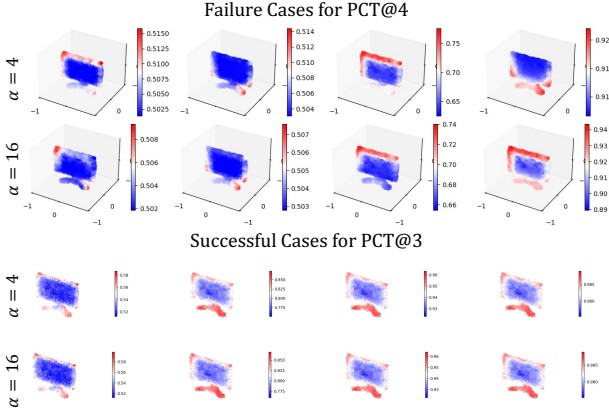

*Figure 14.* Failure cases for PCT@4 with varying $\beta_k \in \{10^1/k, 10^2/k, 10^3/k, 10^4/k\}$, where $k = N \times D$ and $\alpha$ with $D_r = D/\alpha$.

## D.2. Additional Qualitative Results

- In Fig. 15, we visualize the score map of a 'monitor' across eight PC models, together with the critical subsets derived from iteratively dynamic score maps (namely one-pass score maps, 10-iteration score maps/critical subsets, 20-iteration score maps/critical subsets).

- In Fig. 16, we compare InfoCons with dynamic PC-SAM (Zheng et al., 2019) and CP++ (Levi & Gilboa, 2023) on more samples.

- In this link, we provide additional qualitative comparisons following the same settings as Fig. 6 on all test samples

in the 'flower_pot' class of the ModelNet40 dataset across five approaches.

## D.3. Additional Quantitative Results

- In Fig. 17, we additionally demonstrate InfoCons on eight models.

- In Fig. 18, we provide the accuracy degradation of point-drop attack, additionally comparing InfoCons on eight models with PC Saliency Map.

- In Fig. 19, we compare InfoCons on eight models with CP++.

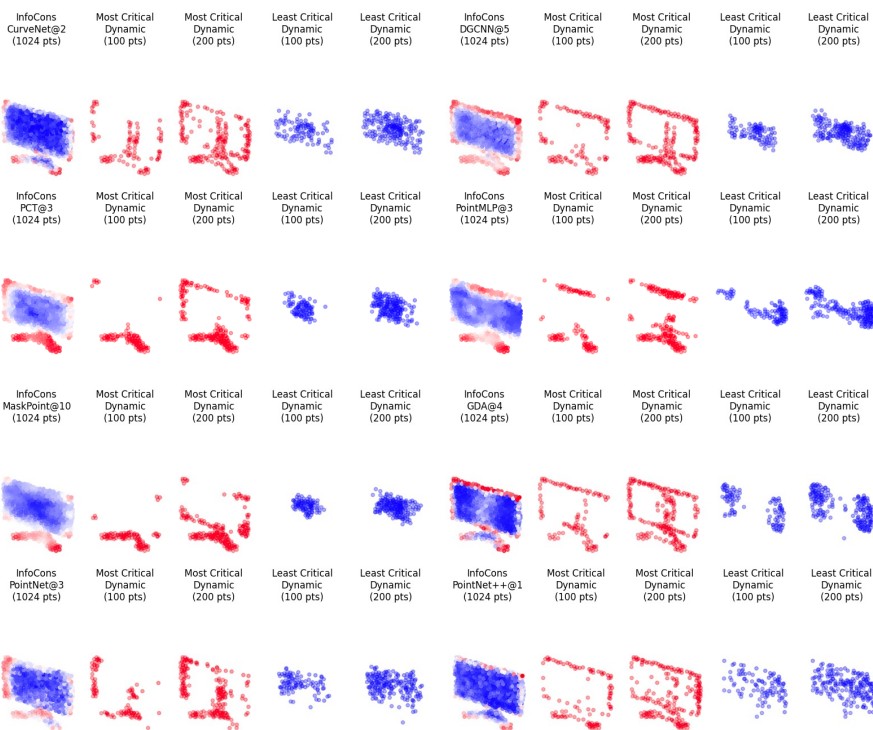

*Figure 15.* InfoCons on the PC labeled as 'monitor' for 8 PC models (from left to right: (i)the 1024 points score map, (ii)the dynamic critical points with 10 iterations and 10 points dropped each iteration, 100 points in total, (iii) dynamic critical points with 20 iteration, 200 points in total, and (iv)-(v) is the least critical points for 10 iterations and 20 iterations).

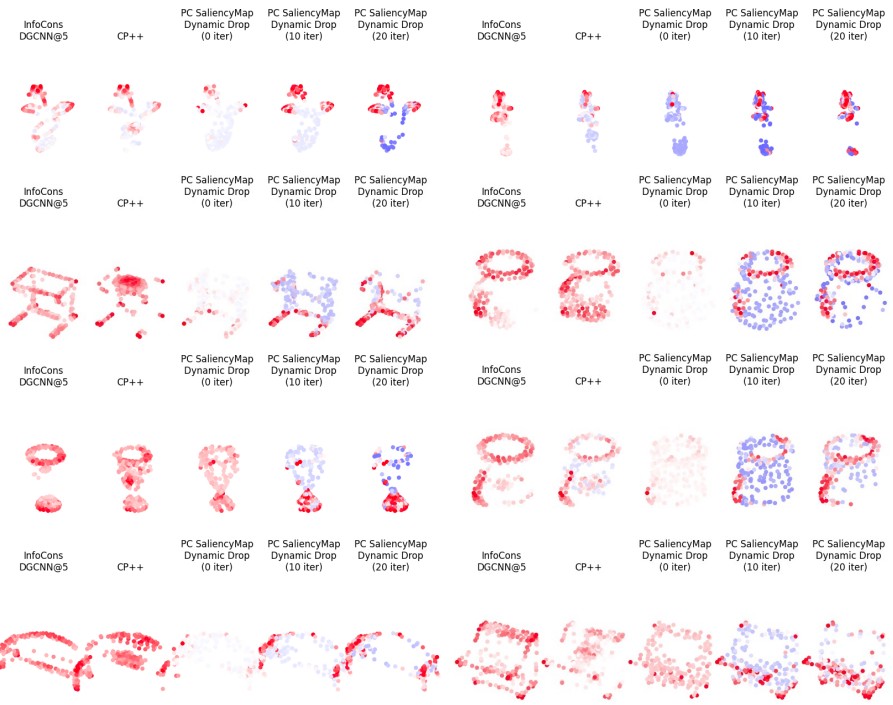

*Figure 16.* InfoCons compared with two baselines on DGCNN (Wang et al., 2019), and we demonstrate 200 points critical subset. From left to right, (i) Info*C*ons, (ii) Critical Points++ (Levi & Gilboa, 2023), (iii)-(v) PC Saliency Map (Zheng et al., 2019) with a dynamic process of *point-dropping-and-then-recalculating-gradient*, and we highlight the dropped points in earlier iterations as high scores.

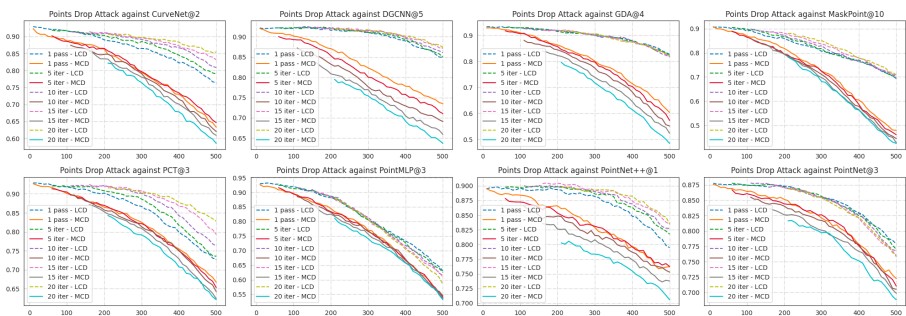

*Figure 17.* Effectiveness evaluation of InfoCons on eight PC models by point-drop attack, where 10 to 500 points are dropped. We report the changes of accuracy under two settings: (i) dropping out the most critical points (MCD) and (ii) dropping out the least critical points (LCD). The accuracy gap between MCD and LCD indicates InfoCons captures the shape-relevant points by assigning them higher scores.

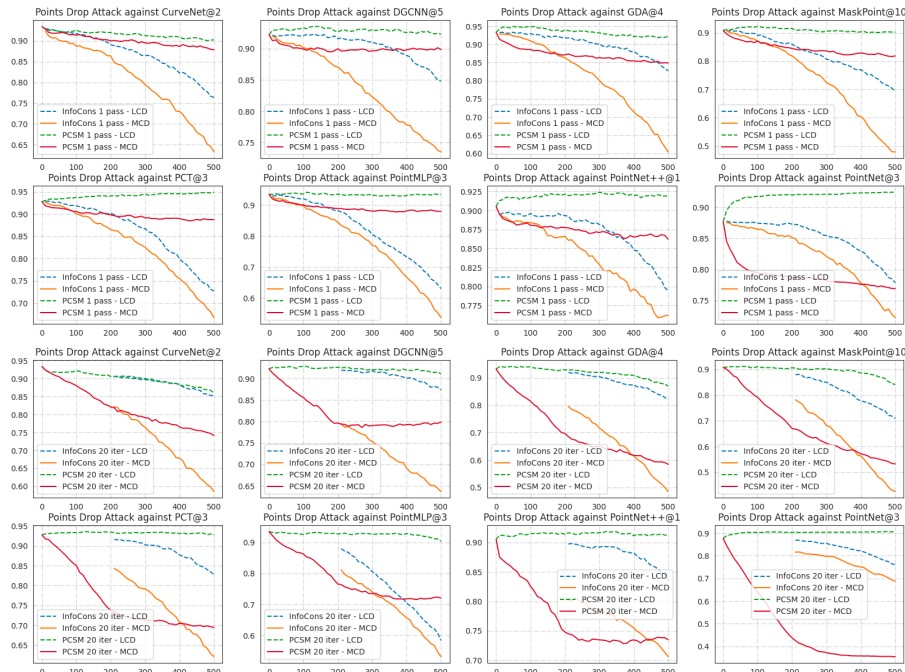

*Figure 18.* Additional comparisons of InfoCons and PC Saliency Map for Fig. 7.

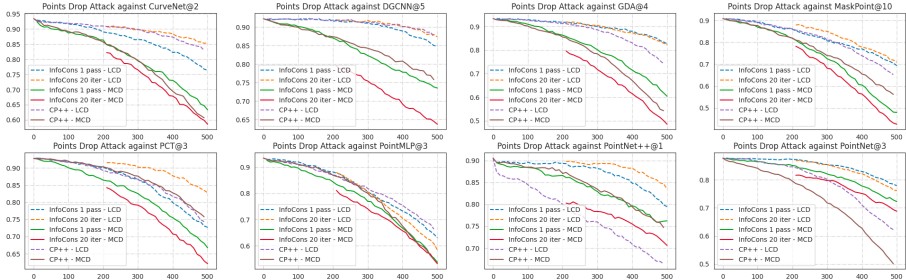

*Figure 19.* Quantitative comparisons between our proposed objectives (Eq. 5 and Eq. 6) and Critical Points++ (Levi & Gilboa, 2023) for eight PC models. We observe an abnormal trend for PointNet++, where CP++ assigns critical points as unimportant, resulting in a rapidly dropping curve for LCD.

