# OpenReview forum: "InfoCons: Identifying Interpretable Critical Concepts in Point Clouds via Information Theory"
_ICML.cc/2025/Conference — ICML 2025 poster_

### Official Review · Reviewer_LVUY · 2025-03-14

**Overall Recommendation:** 2

**Summary:**

The author proposes a novel InfoCons framework based on the principle of information theory. The framework divides point clouds into different 3D concepts with different influences by using the mutual information principle. It also learns meaningful concept structures by combining learnable prior knowledge. The effectiveness of this method is verified on multiple point cloud models.

**Claims And Evidence:**

Yes.

**Essential References Not Discussed:**

No.

**Experimental Designs Or Analyses:**

Yes. The authors evaluated InfoCons on multiple datasets (ModelNet40, ScanObjectNN, KITTI), eight different point cloud models, and two application scenarios.

**Methods And Evaluation Criteria:**

Yes.

**Other Comments Or Suggestions:**

The author needs to check the writing of the entire article. For example, the font of the link in the lower left corner of page 11 is inconsistent.

**Other Strengths And Weaknesses:**

Strengths:
1、The authors proposed a method to solve the interpretability of 3D point cloud models from the perspective of information theory.
2、Extensive experimental evaluations on multiple model architectures, datasets, and applications demonstrate the versatility and effectiveness of InfoCons.
Weakness:
1、The mechanisms behind balancing fidelity and conceptual coherence in explanations remain unclear.
2、The article lacks an introduction to some mathematical symbols.
3、The figure shown by the author is too unclear and the quality of the figure needs to be improved.
4、Although the author used multiple models to verify the effectiveness of the method, these models are too old. It is recommended that the author verify the effectiveness of the model based on the latest method.
5、The Attention Bottleneck lacks some novelty.

**Questions For Authors:**

See Weakness.

**Relation To Broader Scientific Literature:**

The author provides a background introduction to InfoCons from the perspective of point cloud interpretability, information theory, etc. In addition, the author provides a comprehensive review of pooling-based, gradient-based, and black-box query-based methods.

**Theoretical Claims:**

The authors propose an approach inspired by information theory and clearly establish the connection between their goals and theoretical considerations. The mathematical derivations in Sections 3.1-3.3 provide a clear path from theoretical principles to practical implementation.

---

> ### Author Rebuttal · Authors · 2025-03-31
>
> **Dear Reviewer LVUY,**
>
> We sincerely thank you for your thorough review and valuable comments on our paper. We have summarized your concerns into three parts and provided our responses as follows:
>
> ### 1: **Balancing Fidelity and Conceptual Coherence**
>
> Thank you for raising an important question. In our work, we examine the balance between fidelity (faithfulness) and conceptual coherence (interpretability) from two perspectives:
> - (i) Limitations of existing methods: As discussed in lines 68–87 (left part), existing approaches often **lack either conceptual coherence or faithfulness**, which we attribute to the absence of a learnable, unbiased prior.
> - (ii) Challenges in balancing conceptual coherence: When a learnable prior is introduced (Sec. 3.2, line 193), we observe that **feature entanglement among neighboring points** leads to a **conflict between fidelity and conceptual coherence**. To address this issue, in Def. 3.2, we introduce the term $\text{sg}(1-\hat m) \odot  \epsilon$ to compensate for information loss caused by *selecting critical points while excluding their entangled neighbors*. A detailed discussion of this mechanism is provided in lines 201 (right) to 251 (left).
>
> As a result, as shown in Fig. 9, the test accuracy remains stable within a narrow range (92.22% to 92.26%) across different values of $\beta$. It is attributed to the compensation term, which **injects Gaussian noise to replace the influence of unimportant yet entangled points**, effectively mitigating the conflict.
>
> ---
> ### 2: **Clarity of Figures, Mathematical Notation, and Novelty of Attention Bottleneck**
>
> - **Concerns About Notations and Figure Clarity in Fig. 2**:  We acknowledge that some implementation-specific notations (e.g., $\hat{m}, \mu_z, \sigma_z, \epsilon$)  cause clarity issues in Fig. 2-(b). To improve readability, we will relocate these notations to Fig. 5, which will focus on the detailed workflow.
> - **Concerns About the Novelty of Attention Bottleneck**: Attention Bottleneck (AB) is integrated into InfoCons because it can handle data of varying lengths (e.g., different numbers of points). We want to clarify that the main contribution of InfoCons lies in how AB is integrated into our framework. For example, directly applying AB (as a straightforward "selection strategy" in Def. 3.1) is inappropriate for some PC models, as it introduces conflicts between fidelity and conceptual coherence, as mentioned earlier. As presented in Def. 3.2 and Fig. 2-(b) of the current manuscript, we demonstrate how AB (denoted as $\theta$) serves as a component of our IB-based explanation framework, which constitutes our main contribution. We recognize that relocating the lower-left portion of Fig. 2-(b) to Fig. 5 would present our contributions.
>
> ---
> ### 3: **Evaluating on More Recent Point Cloud Models**
>
> We appreciate the reviewer’s suggestion to validate our method on more recent point cloud models. In the current manuscript, we evaluate eight PC models spanning three distinct architecture types: non-hierarchical MLP-based, hierarchical MLP-based, and self-attention-based models (details in Appendix C). We conduct a further survey and find that these models encompass the majority of widely adopted point cloud models in both research and practice, including those used in adversarial attacks.
>
> To further assess our method on a recent model, we conduct a pilot study on Sonata [1] (CVPR 2025 accepted), a newly proposed self-supervised pretraining framework for scene understanding. **Sonata utilizes a novel hierarchical self-attention-based encoder**. Our pilot study extends our method as follows (similar to the object detection experiments in Appendix B.2):
>
> - We follow Sonata’s official setup and re-implement linear probing on the S3DIS dataset for 3D semantic segmentation. Given a test PC, our goal is to explain *why the model incorrectly segments part of a table as a chair*, as shown in subfig-(I) in this link: https://ibb.co/FqXg3tD1
> - The score map based on InfoCons is shown in subfig-(III). Results show that the interaction between the table‘s leg and the chair's seat leads to the incorrect segmentation of the table's leg.
> - More specifically, the implementation of InfoCons (Def.3.2) has made the following adaptations: (i) Replace the sample-wise CE loss with an aggregated point-wise CE loss (i.e., I(C; Y)) over points and target labels of interest, and the information loss term (i.e., I(C; X)) remains unchanged. (ii) To reduce computational overhead, we learn the prior using only the target PC, which contains about a million points.
>
> Finally, regarding the reviewer's suggestions, we will correct the formatting inconsistencies in Appendix B. We appreciate the detailed review and are grateful for your comments. We hope this additional analysis addresses your concerns.
>
>
> [1] Sonata: Self-Supervised Learning of Reliable Point Representations, arxiv25
>
> Best regards,
>
> Authors

---

> > ### Comment · Reviewer_LVUY · 2025-04-06
> >
> > Thank you for the author's response. I have carefully reviewed your reply. Although the author attempted to explain how to balance fidelity and conceptual coherence in the response, the explanation regarding the balance between fidelity and conceptual consistency still does not dispel my concerns. Their explanation relies on empirical observations rather than a reliable theoretical foundation and implementation approach, which may lead to serious issues with the reproducibility and reliability of the method. While the author mentioned preliminary research on Sonata, this remains a small-scale experiment that is insufficient to demonstrate the effectiveness of the method on the latest models (such as existing self-supervised pretraining methods like PointMAE, ReCon, PointGPT, etc.). Additionally, I noticed that Sonata only made its code public in the last week or two, which raises questions about how the authors were able to conduct additional experiments in such a short time. Furthermore, issues with writing, unclear mathematical notation in key formulas, and poor figure quality further exacerbate my concerns about this paper. Therefore, I am adjusting my score to weak reject.

---

> > > ### Author Response · Authors · 2025-04-06
> > >
> > > Thank you for your response. While we appreciate your feedback, we believe that the concerns raised can be addressed.
> > >
> > > ### Q1: Concerns about the Lack of Theoretical Justification and Implementation Details
> > >
> > > - We provide detailed **theoretical justification** and **implementation details** in Sec 3.1 and Sec 3.2, including both mathematical formulation (in Sec. 3) and a feature analysis (Fig. 3).
> > >     (We also thank Reviewers h8Li and vGFq for carefully checking the mathematical notations and confirming that no errors were found. *As a minor clarification, we note that the Theoretical Claims comment refers to "Sections 3.1–3.3", **but our paper does not contain a Section 3.3.***)
> > >
> > > - Our rebuttal aims to offer a more intuitive understanding of the conflict issue, grounded in the theoretical formulation already presented in the paper (with appropriate references).
> > >
> > > - For clarity, we briefly restate the formulation here: Motivated by the need to develop a "selective strategy" for identifying critical points, we introduce a soft mask $\hat m$ and compute the bottlenecked feature $\hat z=\hat m\odot z(x)$ (Eq.4, Def.3.1). To address the issue of entangled features among neighboring of critical points, we reformulate it as $\hat z= \hat m\odot z(x)+\text{stop-gradient}(1-\hat m)\odot \epsilon$ (Eq.6, Def.3.2), where $\epsilon$ denotes a D-dimentional Gaussian variable, under the assumption that the features of individual points are i.i.d. (line 206, right column). **The Feature analysis in Figure 3 empirically supprots this assumption**, as the distributions of three randonly sampled point features appear arrpximately i.i.d.
> > >
> > > - Thus, when we maximize $\log q(y|\hat z)$ and minimize $KL(\hat z||q(\hat z))$ (with $q(\cdot)$ being a Gaussian prior), the compensation term $\text{stop-gradient}(1-\hat m)\odot \epsilon$ maintains the overall distribution of $\hat z$, while $\hat m\odot z(x)$ effectively selects critical points and corrupts the features of unimportant points.
> > >
> > > We will open-source learned weights and scripts for reproducibility.
> > >
> > > ### Q2: Concerns about *small-scale experiments on the latest models*
> > >
> > > We would like to clarify three key points regarding the experiments related to the latest models and SSL-based pretraining methods:
> > >
> > > - Firstly, our method is primarily designed for **point cloud classification in the supervised learning** (since our objective depends on $I(C;Y)$, which requires label supervision), rather than self-supervised setting. Therefore, we have conducted comprehensive experiments including eight PC classification models.
> > >
> > > - Second, we have included MaskPoint (ECCV 2022), a self-supervised pretraining method, which is concurrent with PointMAE (ECCV 2022).
> > > Notably, MaskPoint for ModelNet40 classification follows a typical SSL pipeline, where the pretrained encoder is frozen, and a downstream classification head is fine-tuned. Thus, the inclusion of MaskPoint already covers the representative paradigm of SSL-based models.
> > >
> > > - Third, we note that ReCon (ICML 2023) is pretrained on multi-modal data (e.g., images and/or texts), and PointGPT (NeurIPS 2023) adopts a generative pretraining objective. And both methods follow the typical SSL pipeline for classification.
> > >
> > > We appreciate the reviewer’s insightful suggestion. Systematic experiments on SSL-pretrained models will be considered as part of our future work, with extensions to **multi-modal setting** and a focus on **generalization ability**.
> > >
> > >
> > > ### Q3: Concerns about *conducting additional experiments in such a short time*
> > > - Sonata's open-sourced weights became publicly available on March 20. After reviewing their paper (also publicly available on March 20), we used their script to re-implement the method (which was available on March 21).
> > > - Given our experience in re-implementing at least eight point cloud models (as demonstrated in our paper), we are confident in our ability to conduct a pilot study within the rebuttal period.
> > >
> > > ### Q4: Issues with Writing, Unclear Mathematical Notation in Key Formulas, and Poor Figure Quality
> > > Firstly, regarding the **writing issues with the URL format in footnotes in Appendix B.2**, we acknowledge that the format for the last two links may not be consistent. However, this does not affect the essential information. We can easily correct this by inserting the \url{} command.
> > >
> > > Additionally, we believe the comments about some weaknesses **lack specificity**.
> > > - We kindly ask the reviewer to clearly point out **any other writing issues in our manuscript**, and we will be happy to address them immediately.
> > > - We also request the reviewer to specify **which mathematical notations in particular equations, figures, or sections of the main text are unclear**, so that we can make the necessary clarifications.
> > > - Finally, we would appreciate it if the reviewer could identify **which figure(s) or if all figures are of poor quality**, and we will gladly make the required improvements.
> > >
> > > Best regards,
> > >
> > > Authors

---

### Official Review · Reviewer_vGFq · 2025-03-14

**Overall Recommendation:** 3

**Summary:**

This paper mainly focuses on how to extract interpretable key concepts in point cloud models to enhance the interpretability of the models. This work addresses the issue that existing methods often fail to simultaneously meet the two criteria of "faithfulness" and "conceptual cohesion" when providing interpretable subsets. It proposes a framework named InfoCons based on information theory. By maximizing the mutual information between the key subsets and the model decisions, it ensures faithfulness. Meanwhile, it introduces a learning unbiased prior to minimize the mutual information between the key subsets and the input point cloud, thereby encouraging the formation of meaningful conceptual structures. A large number of experiments have verified the effectiveness of the method.

**Claims And Evidence:**

Most of the author's propositions are supported by clear quantitative or qualitative analyses. However, regarding the starting point of this article, "an ideal critical subset should be faithful (preserving points that causally influence predictions) and conceptual coherent
(forming semantically meaningful structures that align with human perception)". This claim seems to have no explanation provided for it. I'm not quite sure whether this is a definition that others have proved or is a widly-used consensus, or a definition put forward by the author in this article. I think this statement should be substantiated and explained.)

**Essential References Not Discussed:**

There is no obvious lack of references.

**Experimental Designs Or Analyses:**

In Section 4.1, the qualitative analysis experiment conducted by the author merely relied on the performance of different methods on a few small samples to demonstrate the validity of his/her method. I think this experiment seems rather insufficiently rigorous and should have more or the entire dataset performance as supplementary evidence.

**Methods And Evaluation Criteria:**

The method InfoCons proposed from the perspective of information theory is useful.

**Other Comments Or Suggestions:**

no comments

**Other Strengths And Weaknesses:**

I think the merit of this paper lies in its analysis of model interpretability from the perspective of information theory, which provides certain theoretical basis. Meanwhile, the paper has conducted numerous experiments to prove the effectiveness of the proposed method. Regarding the drawbacks, at present, one aspect is the aforementioned concerns about the initial statement "an ideal critical subset should be faithful and conceptually coherent" and the qualitative analysis.

**Questions For Authors:**

see Other Strengths And Weaknesses

**Relation To Broader Scientific Literature:**

This paper mainly applies information theory to conduct interpretable analysis on point clouds. I believe this idea can also be further extended to other research fields.

**Theoretical Claims:**

I have checked the author's theoretical claims. In my opinion, they are correct, especially in the section about the deep variational information bottleneck.

---

> ### Author Rebuttal · Authors · 2025-03-31
>
> **Dear Reviewer vGFq,**
>
> We sincerely thank you for your thorough review and detailed comments on our paper.
> In response to your concerns regarding the statement on “Good Explanations” and our qualitative comparison, we provide the following clarifications.
>
> ### 1: **Clarification on “Good Explanations”**
>
>
> Our definition of *what constitutes a good explanation for point cloud models* builds upon **widely accepted principles** in interpretability research, which we clarify as follows:
>
> - (i) The desiderata of explanation methods, **faithfulness and interpretability**, date back to LIME [1] (KDD 2016) and have since been widely adopted in subsequent interpretability methods (e.g., LIME3D), as stated in the abstract of the LIME paper:
> > "...a novel explanation technique that explains the predictions of any classifier **in an interpretable and faithful manner** ...".
>
> - (ii) In our work, we propose a **statement on a good critical subset** (i.e., an ideal critical subset should be faithful to model predictions and conceptually coherent with human prior, as detailed in line 67, left) in the context of **critical-point-based explanations for point cloud data**.
> This aligns with the notion used in Grad-CAM [2] (IJCV 2019). Grad-CAM also introduced a widely accepted principle for *what constitutes a good visual explanation* in *image classification* (page 2 of the Grad-CAM paper):
> > "**What makes a good visual explanation?** Consider image classification – a ‘good’ visual explanation from the model for justifying any target category should be (a) class-discriminative (i.e. localize the category in the image) and (b) high-resolution (i.e. capture fine-grained detail)."
>
>     However, the process of explaining image classification is significantly different from that of explaining point cloud data. For example, *capturing fine-grained detail* in an image often implies a clear boundary between objects of interest and unimportant background pixels, whereas such a boundary may not exist in point cloud data, which consists of thousands of unordered points (as we discussed in the right part of line 60).
> - (iii) We therefore adopt the term *conceptual coherence*, drawing from concept-level explanations [3-5], where human-understandable concepts provide a more general formulation of interpretability.
> In the image domain, concept-level explanations can be attributes labeled by humans [3] or selective patches [4, 5].
> In our work, **critical concepts** can be intuitively described as selective **sub-PCs** that frequently appear in the training dataset and contribute significantly to corresponding labels. Here, we introduce two key adaptations: (i) The human prior is replaced with a learnable prior derived from the training dataset. (ii) Critical concepts are explicitly extracted/selected from the original PC.
>
> To summarize, our claim is consistent with these widely accepted principles and appropriately formulated for point cloud data. We appreciate the reviewer’s suggestion and will include additional discussion in the appendix to clarify this point.
>
> ---
>
> [1] "Why should I trust you?" Explaining the predictions of any classifier. KDD 2016.
>
> [2] Grad-CAM: Visual Explanations from Deep Networks via Gradient-Based Localization. IJCV 2019.
>
> [3] Concept bottleneck models. ICML 2020.
>
> [4] Approximating cnns with bag-of-local-features models works surprisingly well on imagenet. ICLR 2019.
>
> [5] Explaining generalization power of a dnn using interactive concepts. AAAI 2024.
>
> ---
> ### 2: **Additional Qualitative Analyses**
>
> In addition to Fig. 6 in Sec. 4.1, we have provided further qualitative comparisons in Appendix-Fig. 14 in the current manuscript.
>
> Furthermore, in response to the reviewer’s suggestion, we conduct additional qualitative comparisons following the same settings as Sec. 4.1 on 20 samples (i.e., all test samples in the 'flower_pot' class of the ModelNet40 dataset) across five approaches. The results are available at this link: https://ibb.co/zhG7KGQs (this may take some time to load due to the image size). We will include these additional examples in Appendix D.1 (Additional Qualitative Results).
>
> Additionally, since our method provides score maps for explanations, we have already conducted quantitative evaluations over the entire test dataset to rigorously assess its effectiveness. Specifically, we have conducted drop attack (Sec. 4.2), adversarial attack (Tab. 1), and data augmentation (Tab. 2) experiments. These quantitative evaluations provide measurable evidence of our method’s effectiveness.
>
> We sincerely appreciate your insightful suggestions and will incorporate the additional qualitative results into our paper.
>
> Best regards,
>
> Authors

---

### Official Review · Reviewer_h8Li · 2025-03-21

**Overall Recommendation:** 4

**Summary:**

This paper addresses the problem of explaining decisions made by point cloud classification models, which is particularly important in applications such as autonomous vehicles. Current methods mainly focus on mathematical values like gradients or neuron activations. However, the authors break down the 3D point cloud into concepts that are more understandable to humans, allowing for a better evaluation of how the data influences the decision from a human perspective. The authors propose a novel framework, InfoCons, that applies information-theoretic principles to decompose a point cloud into 3D concepts with varying levels of influence on model predictions. The critical subset is determined by the most discriminative concept. The proposed method, InfoCons, attributes model predictions to key concepts, offering more precise and less redundant explanations. Comparative experiments show that InfoCons effectively identifies significant points in point clouds.

**Claims And Evidence:**

The claims made in the submitted paper are supported by clear and convincing evidence. The authors conduct comprehensive experiments. The example point clouds are selected from the synthetic dataset ModelNet40 and two real-world datasets: ScanObjectNN for shape classification and KITTI for object detection. They compare their model with four baselines: Critical Points, PC Saliency Map, Critical Points++, and LIME3D. Additionally, they perform comparative experiments on PC models: PointNet, CurveNet, GDA, PointMLP, DGCNN, Maskpoint, and PCT.
The qualitative comparison in Figure 6 is helpful, showing how InfoCons assigns importance to critical points in point clouds, which is key for interpretability in applications like autonomous vehicles. They also perform Adversarial Attack, demonstrating how InfoCons compares to other methods. Additionally, the authors provide useful insights on the key hyperparameters of their model.

**Essential References Not Discussed:**

I am not certain about any specific essential works that should have been included but were not.

**Experimental Designs Or Analyses:**

The authors logically present their experiments, providing clear and detailed descriptions. They use multiple datasets, various baseline methods, and evaluate the results from different perspectives. The obtained results are supported not only by metric values in tables but also by visualizations in graphs. Based on the presented findings, it can be concluded that the method has been thoroughly tested.

**Methods And Evaluation Criteria:**

The authors propose appropriate methods and evaluation criteria. They use benchmark datasets, both synthetic and real-world. In addition, they compare their approach with baseline methods. They also utilize various metrics such as ASR, CD, and HD.

**Other Comments Or Suggestions:**

No additional comments or suggestions.

**Other Strengths And Weaknesses:**

Strengths:
- The problem is well-described and well-justified
- A new method is proposed that takes into account additional factors, such as human-eye interpretability
- Detailed experiments are conducted
- Justification for the use of the work, such as its application to autonomous driving
- Key hyperparameters are clearly presented

Weaknesses:
- The related work section is too brief
- The structure is a bit unusual. I would suggest placing the related works section at the beginning, perhaps after the introduction, rather than after the experimental results
- Figure 2 is not very clear to me; it’s easier to understand the method from the description than from the figure itself

**Questions For Authors:**

I do not have any questions for the authors

**Relation To Broader Scientific Literature:**

The paper’s contributions build upon existing point cloud explanation methods, such as Critical Points and Critical Points++ . It also addresses the bias in gradient-based methods like PC Saliency Map  and improves upon query-based methods like LIME3D by providing more interpretable critical point selection. Overall, the paper enhances existing approaches, offering more accurate and conceptually meaningful interpretations of point cloud models.

**Theoretical Claims:**

In the methodology chapter, the authors present theoretical claims and mathematical formulas. In particular, they refer to the Information Bottleneck (IB) approach for critical points. I have reviewed the mathematical formulas and their notations, and I found no errors. However, some of the introductions and explanations of the theoretical concepts could have been more clearly presented.

---

> ### Author Rebuttal · Authors · 2025-03-31
>
> **Dear Reviewer h8Li,**
>
> We sincerely appreciate your thorough review and your positive assessment of our work. Based on your feedback, we will refine the writing and presentation to enhance readability as follows:
>
> ### **1: Expanding the Related Work Section and Adjusting Paper Structure**
> In the current manuscript, we have integrated methodological comparisons with closely related approaches (e.g., CP, CP++, and PCSAM) within Sec. 3.1 (line 201) and Sec. 3.2 (line 252) to maintain narrative flow while avoiding redundancy.
> However, we acknowledge that an overview of point cloud interpretability literature in Sec. 2 would further benefit readers. To address this, we will
> - (i) **Introduce a new subsection in the Background section (Sec. 2)** that specifically discusses prior **attribution methods for point cloud models**, including CP, CP++, and PCSAM.
> - (ii) **Expand the discussion of these methods in a new appendix section** to provide a comprehensive review of existing interpretability methods for point cloud models, considering their architectural characteristics.
>
> ---
>
> ### **2: Clarification of Figure 2 for Improved Understanding**
>
> Thank you for pointing out the clarity issue with Fig. 2.
> To improve its readability, we will revise the lower-left portion of Fig. 2-(b) to emphasize the high-level framework.
> We will remove implementation-specific notations (e.g., $\hat{m}, \mu_z, \sigma_z, \epsilon$) from Fig. 2-(b).
> After formally introducing these notations in Eq. 5 and Eq. 6 on page 4, we will present the implementation-specific details (i.e., $\hat{m}$) in Fig. 5 to clarify the overall data flow.
> This revision will ensure that Fig. 2 centers on the high-level framework, while Fig. 5 provides a more detailed view.
>
> We greatly appreciate your time and valuable feedback, which will significantly enhance the clarity and presentation of our paper.
>
> Best regards,
>
> Authors

---

### Decision · Program_Chairs · 2025-05-01

**Decision:**

Accept (poster)

**Comment:**

This paper proposes a novel interpretability method for point clouds grounded in information-theoretic principles. The contribution is timely and relevant, addressing a less-explored yet important aspect of 3D deep learning. The overall reception from the reviewers is positive, with appreciation for the originality of the approach and the soundness of the experimental protocol.

The authors provide a consistent evaluation aligned with their methodology, and the experiments are generally well-conducted. Reviewers raised concerns regarding the inclusion of certain important baselines—particularly Sonata—which were appropriately addressed during the rebuttal phase. This responsiveness is commendable and strengthens the empirical foundation of the work.

However, one of the key remaining concerns, raised by reviewer LVUY, is the limited scope of model applicability. Specifically, the effectiveness of the proposed method is not demonstrated on more recent or widely-used self-supervised models such as PointMAE, ReCon, or PointGPT. This raises questions about the generalizability and practical relevance of the method in contemporary settings.

Additionally, reviewers noted issues with presentation clarity, which affects the accessibility of the paper. Some parts of the text require refinement to improve the overall readability and ease of understanding.

In summary, the work presents a promising and original contribution to the interpretability of point cloud models. While there are areas for improvement—particularly in presentation and broader applicability—the core idea is compelling, and the authors have demonstrated a willingness to engage with reviewer feedback.